Integrating whole genome sequencing and machine learning for predicting antimicrobial resistance in critical pathogens: a systematic review of antimicrobial susceptibility tests

Ardila Carlos M. martin.ardila@udea.edu.co 1 2
Yadalam Pradeep K. 3
González-Arroyave Daniel 4
1 Basic Sciences Department, Faculty of Dentistry, Universidad de Antioquia , Medellin , Colombia
2 CIFE University Center , Cuernavaca , Mexico
3 Periodontics, Saveetha University , Saveetha , India
4 Surgery, Universidad Pontificia Bolivariana , Medellin , Colombia
Uversky Vladimir
Electronic publication date: 2024 Oct 9
Publication date: 2024
Volume: 12
Electronic Location ID: e18213
Received 2024 Jul 15; Accepted 2024 Sep 11
Copyright: ©2024 Ardila et al.
Copyright year: 2024
Copyright holder: Ardila et al.
License: This is an open access article distributed under the terms of the Creative Commons Attribution License, which permits unrestricted use, distribution, reproduction and adaptation in any medium and for any purpose provided that it is properly attributed. For attribution, the original author(s), title, publication source (PeerJ) and either DOI or URL of the article must be cited.
License URL: https://creativecommons.org/licenses/by/4.0/

Keywords: Antimicrobial resistance, Machine learning, Whole genome sequencing, Prediction models, Risk score

Funding: The authors received no funding for this work.

==============================
Background

Infections caused by antibiotic-resistant bacteria pose a major challenge to modern healthcare. This systematic review evaluates the efficacy of machine learning (ML) approaches in predicting antimicrobial resistance (AMR) in critical pathogens (CP), considering Whole Genome Sequencing (WGS) and antimicrobial susceptibility testing (AST).

Methods

The search covered databases including PubMed/MEDLINE, EMBASE, Web of Science, SCOPUS, and SCIELO, from their inception until June 2024. The review protocol was officially registered on PROSPERO (CRD42024543099).

Results

The review included 26 papers, analyzing data from 104,141 microbial samples. Random Forest (RF), XGBoost, and logistic regression (LR) emerged as the top-performing models, with mean Area Under the Receiver Operating Characteristic (AUC) values of 0.89, 0.87, and 0.87, respectively. RF showed superior performance with AUC values ranging from 0.66 to 0.97, while XGBoost and LR showed similar performance with AUC values ranging from 0.83 to 0.91 and 0.76 to 0.96, respectively. Most studies indicate that integrating WGS and AST data into ML models enhances predictive performance, improves antibiotic stewardship, and provides valuable clinical decision support. ML shows significant promise for predicting AMR by integrating WGS and AST data in CP. Standardized guidelines are needed to ensure consistency in future research.

Introduction

Antimicrobial resistance (AMR) is the ability of bacteria to withstand antimicrobial treatments, particularly antibiotics. Infections caused by antibiotic-resistant bacteria are a significant concern for modern healthcare, posing a serious public health risk (Ahmad et al., 2023). Projections estimate that bacterial infections could result in approximately 10 million deaths annually by 2050 (Lüftinger et al., 2023). A recent meta-analysis on the impact of resistant bacteria on human health revealed that in 2019, antibiotic-resistant bacteria (ARBs) directly caused 1.27 million deaths, with an additional 4.95 million deaths associated with ARBs (Antimicrobial Resistance Collaborators, 2022). Moreover, ARBs are identified as a leading cause of mortality in low-income countries (Antimicrobial Resistance Collaborators, 2022; Ruiz-Blanco et al., 2022). Infections with Escherichia coli, Staphylococcus aureus, Klebsiella pneumoniae, Acinetobacter baumannii, and Pseudomonas aeruginosa, all classified as critical and high-priority pathogens (CP) by the World Health Organization (WHO) (Ahmad et al., 2023; Antimicrobial Resistance Collaborators, 2022), account for a significant portion of these deaths. In 2019, over 100,000 deaths were attributed to antimicrobial resistance (AMR) caused by a single pathogen-drug combination: methicillin-resistant S. aureus (MRSA). Six other combinations caused 50,000 to 100,000 deaths each, including multidrug-resistant E. coli, fluoroquinolone-resistant E. coli, carbapenem-resistant A. baumannii, carbapenem-resistant K. pneumoniae, and third-generation cephalosporin-resistant K. pneumoniae. Additionally, a recent assessment of the clinical pipeline revealed the development of 50 antibiotics, but only 12 showed efficacy against certain priority Gram-negative bacteria (Butler & Paterson, 2020; Pormohammad, Nasiri & Azimi, 2019).

Research indicates that the rapid administration of appropriate antimicrobials significantly improves patient outcomes. For instance, in cases of bacteremia, the risk of death doubles if effective antibiotics are not administered within 24 h. Globally, only about half of antibiotic prescriptions are accurate. Consequently, quick point-of-care diagnostic tests are crucial for addressing this issue (Ahmad et al., 2023; Milani et al., 2019).

The current culture-based methods for detecting and diagnosing pathogenic diseases are insufficient. Most culturable bacteria associated with diseases can be detected after 24-48 h of incubation. Additionally, pathogen identification often requires an extra 2-4 h, and if AMR is suspected, antibiotic susceptibility testing (AST) adds another 18-24 h. Consequently, the total time to collect patient samples and obtain information on antibiotic susceptibility patterns in clinical practice can range from 2 to 4 days at best (Ahmad et al., 2023; Taxt et al., 2020).

Emerging micro- and nanotechnologies for bacterial identification and AST include phenotypic methods like microfluidic-based bacterial culture, and molecular techniques such as multiplex PCR, hybridization probes, nanoparticles, synthetic biology, and mass spectrometry. While PCR and mass spectrometry have improved bacterial detection in positive cultures, they have limitations. PCR requires a predetermined target, and MALDI-TOF mass spectrometry is costly (Ahmad et al., 2023; Taxt et al., 2020; Humphries et al., 2023). Additionally, these methods, which assess one or more resistance genes, are inadequate for predicting antimicrobial susceptibility because resistance often results from a complex interplay of resistance genes, regulatory factors, and mutations that together produce a phenotypic susceptibility profile (Humphries et al., 2023).

Whole genome sequencing (WGS) offers a solution to some of these challenges by obviating the requirement for specialized primers or probes. Additionally, with the increasing affordability of real-time sequencing, WGS has emerged as a feasible alternative to the laborious, culture-dependent methods of the past (Ahmad et al., 2023). Moreover, genome sequencing data offer an additional dimension to AMR research, enabling the analysis of genetic pathways underlying AMR in individual strains (Su, Satola & Read, 2019; VanOeffelen et al., 2021). Several publicly accessible services have been established to assist in identifying AMR indications based on the presence of resistance-associated single nucleotide polymorphisms (SNPs) and genes (VanOeffelen et al., 2021; Bortolaia et al., 2020). Integration of AST data with genome sequences also holds promise in uncovering genomic regions directly involved in resistance, influenced by epistasis, or linked to the emergence of AMR (VanOeffelen et al., 2021; Hendriksen et al., 2019).

The utilization of machine learning (ML) methods for forecasting AMR indicators and pinpointing genetic regions associated with resistance has garnered considerable interest in recent literature (VanOeffelen et al., 2021; Anahtar, Yang & Kanjilal, 2021). ML technologies leverage a wide array of variables inherent in genomic data to construct nonlinear models that forecast phenotypic AST outcomes (Humphries et al., 2023). ML offers an alternative method for predicting AMR from sequence data without necessitating prior knowledge of chromosomal alterations or mobilizable genes (Aytan-Aktug et al., 2020; Moradigaravand et al., 2018a). Numerous ML approaches can consider the implications of multiple mutations and/or mobilizable genes. Various studies have employed different ML techniques to forecast AMR profiles for diverse bacterial species and drug combinations (Aytan-Aktug et al., 2020; Moradigaravand et al., 2018a; Drouin et al., 2019). The primary distinction among these studies lies in how bacterial genomes are transformed into features, which are subsequently input into ML algorithms (Aytan-Aktug et al., 2020). Unlike culture-based AST or nucleic acid amplification tests, which are frequently constrained in the scope of resistant phenotypes ascertainable through a single test, WGS-AST enables the simultaneous determination of antibiotic resistance phenotypes across the entirety of the genome. Furthermore, it facilitates the screening of phenotypes influenced by multiple loci with ease (Ahmad et al., 2023; Lüftinger et al., 2023; Humphries et al., 2023; Su, Satola & Read, 2019; VanOeffelen et al., 2021). However, no systematic review has been identified that has evaluated the ability of ML to predict antimicrobial resistance in CP using WGS. Therefore, the objective of this systematic review is to evaluate the efficacy of ML approaches in predicting AMR in CP, considering WGS-AST.

Materials and Methods

Protocol and registration

The systematic review followed a search methodology in line with PRISMA (Preferred Reporting Items for Systematic Reviews and Meta-analyses) guidelines (Page et al., 2021). The review protocol was officially registered on PROSPERO.

Eligibility criteria

The systematic review was guided by a question formulated within the Population, Intervention, Comparison, and Outcomes (PICO) framework:

P: samples of patients with CP subjected to AST.

I: Machine learning approaches utilizing WGS-AST

C: Alternative prediction approaches.

O: Prediction of AMR.

This review included studies evaluating the efficacy of ML in predicting AMR in CP, utilizing WGS data and AST. Exclusion criteria comprised animal and in vitro studies, case series and case reports. Moreover, reviews, brief communications, conference proceedings, abstracts, and studies lacking essential information about ML s and predicted performance indicators were omitted.

Information sources

The search strategy involved reviewing various databases, including PubMed/MEDLINE, Web of Science, EMBASE, SCOPUS, and SCIELO, along with searching gray literature sources via Google Scholar. A comprehensive electronic database search was conducted from the inception of these databases until June 2024, without any language restrictions. Moreover, further records were identified by examining the reference lists and citations of all selected full-text papers for potential inclusion in this study.

Search strategy

The search included the following terms: “whole genome sequencing” AND “microbiome” AND “genomics” AND “genome” AND “antibiotic resistance genes” AND “antimicrobial resistance prediction “AND “disk diffusion antimicrobial tests” AND “agar dilution” AND “minimal inhibitory concentration” AND “antimicrobial susceptibility testing” AND “antimicrobial resistance” OR “antibiotic resistance” AND “microbial” OR “bacterial” AND “Escherichia coli” AND “Staphylococcus aureus” AND “Klebsiella pneumoniae” AND “Acinetobacter baumannii” AND “Pseudomonas aeruginosa” AND “infection” AND “machine learning” OR “ machine learning algorithms “OR “deep learning” OR “prediction model” OR “risk assessment” OR “risk prediction”. These search methods employ database-specific syntax and operators to retrieve articles related to the provided queries. Adjustments can be required to adapt the single exploration functionality and syntax rules of each database. Table 1 displays the search strategies for each specified database using the provided terms.

Table 1 Explorations managed in the selected databases.

Database	Search strategy	
PubMed/MEDLINE	((“whole genome sequencing” AND “microbiome” AND “genomics” AND “genome”) AND (“antibiotic resistance genes” AND “antimicrobial resistance prediction “AND “disk diffusion antimicrobial tests” AND “agar dilution” AND “minimal inhibitory concentration” AND “antimicrobial susceptibility testing”) AND (”antimicrobial resistance” OR ”antibiotic resistance”) AND (”microbial” OR ”bacterial”) AND ”Escherichia coli ” AND “Staphylococcus aureus” AND “Klebsiella pneumoniae” AND “Acinetobacter baumannii” AND “Pseudomonas aeruginosa” AND ”infection” AND (”machine learning” OR“machine learning algorithms” OR ”deep learning” OR ”prediction model” OR “risk assessment” OR “risk prediction”))	
Scopus	TITLE-ABS-KEY((“whole genome sequencing” AND “microbiome” AND “genomics” AND “genome”) AND (“antibiotic resistance genes” AND “antimicrobial resistance prediction “AND “disk diffusion antimicrobial tests” AND “agar dilution” AND “minimal inhibitory concentration” AND “antimicrobial susceptibility testing”) (”antimicrobial resistance” OR ”antibiotic resistance”) AND (”microbial” OR ”bacterial”) AND ”Escherichia coli ” AND “Staphylococcus aureus” AND “Klebsiella pneumoniae” AND “Acinetobacter baumannii” AND “Pseudomonas aeruginosa” AND ”infection” AND (”machine learning” OR “machine learning algorithms” OR ”deep learning” OR ”prediction model” OR “risk assessment” OR “risk prediction”))	
Scielo	(“whole genome sequencing” AND “microbiome” AND” “genomics” AND “genome”) AND (“antibiotic resistance genes” AND “antimicrobial resistance prediction “AND “disk diffusion antimicrobial tests” AND “agar dilution” AND “minimal inhibitory concentration” AND “antimicrobial susceptibility testing”) AND (”Antimicrobial resistance” OR ”antibiotic resistance”) AND (”microbial” OR ”bacterial”) AND ”Escherichia coli ” AND “Staphylococcus aureus” AND “Klebsiella pneumoniae” AND “Acinetobacter baumannii” AND “Pseudomonas aeruginosa” AND ”infection” AND (”machine learning” OR “machine learning algorithms” OR ”deep learning” OR ”prediction model” OR “risk assessment” OR “risk prediction”)	
Embase	(“whole genome sequencing” AND “microbiome” AND“genomics” AND “genome”) AND (“antibiotic resistance genes” AND “antimicrobial resistance prediction “AND “disk diffusion antimicrobial tests” AND “agar dilution” AND “minimal inhibitory concentration” “antimicrobial susceptibility testing”) AND (“antimicrobial resistance’ OR ’antibiotic resistance’) AND (’microbial’ OR ’bacterial’) AND ’Escherichia coli’ AND ’Staphylococcus aureus’ AND ’Klebsiella pneumoniae’ AND ’Acinetobacter baumannii’ AND ’Pseudomonas aeruginosa’ AND ’infection’ AND (’machine learning’ OR “machine learning algorithms” OR ’deep learning’ OR ’prediction model’ OR “risk assessment” OR “risk prediction”)	
Web of Science	TS=(“whole genome sequencing” AND “microbiome” AND ““genomics” AND “genome”) AND (“antibiotic resistance genes” AND “antimicrobial resistance prediction “AND “disk diffusion antimicrobial tests” AND “agar dilution” AND “minimal inhibitory concentration” “antimicrobial susceptibility testing”) AND TS=(”Antimicrobial resistance” OR ”antibiotic resistance”) AND TS=(”microbial” OR ”bacterial”) AND TS=”Escherichia coli ” AND “Staphylococcus aureus” AND “Klebsiella pneumoniae” AND “Acinetobacter baumannii” AND “Pseudomonas aeruginosa” AND TS=”infection” AND TS=(”machine learning” OR “machine learning algorithms” OR ”deep learning” OR ”prediction model” OR “risk assessment” OR “risk prediction”)	
Google Scholar	“whole genome sequencing” AND “microbiome” AND “genomics” AND “genome”AND “antibiotic resistance genes” AND “antimicrobial resistance prediction “AND “disk diffusion antimicrobial tests” AND “agar dilution” AND “minimal inhibitory concentration”AND “antimicrobial susceptibility testing” AND ”Antimicrobial resistance” OR ”antibiotic resistance” AND ”microbial” OR ”bacterial” AND ”Escherichia coli” “Staphylococcus aureus” AND “Klebsiella pneumoniae” AND “Acinetobacter baumannii” AND “Pseudomonas aeruginosa” AND ”infection” AND ”machine learning” OR “machine learning algorithms” OR ”deep learning” OR ”prediction model” OR “risk assessment” OR “risk prediction”	

Study selection

Two investigators (CMA and DGA) individually assessed the eligibility of titles and abstracts, followed by a comprehensive review of full-text studies. Full-text evaluation was conducted independently to ascertain eligibility. When discrepancies arose, they were initially discussed between the two investigators to reach a consensus. If disagreements persisted after discussion, a third scholar (PKY) was consulted to provide an independent evaluation and make the final decision. Interobserver concordance was evaluated by means of the Kappa test for statistical significance, with a threshold of >90 indicating consistency.

Data collection

Two researchers (CMA and DGA) autonomously extracted information applying tailored data extraction strategies. A comparative study was managed to ensure consistency in the acquired information. The resistance report, variables utilized, machine learning approach, performance measures, and WGS data for model construction and validation were gathered from the revised articles. Furthermore, an orderly recording of points including authors, publishing year and country was carried out.

Assessment of bias risk and study quality in individual studies

The PROBAST instrument, which assesses both the risk of bias and the applicability of prediction model research for systematic reviews, was employed to evaluate bias (Moons et al., 2019). A total of 20 signaling items were examined across four domains: participants, predictors, results, and analysis. Moreover, the first three domains were evaluated for each involved investigation. The risk of bias was categorized as “high risk” if at least one question was answered “no” or “probably no” without suitable justification. A field was contemplated to have an “unclear risk” if indispensable report for some signaling item was absent, but there were no items that would classify the domain as high risk.

Summary measurements

Data from the included studies were collected using descriptive statistics such as mean differences, standard deviation values, and ranges, with a focus on continuous outcomes. If the papers demonstrated significant homogeneity, the possibility of managing a meta-analysis was assessed.

Results

Study selection

After searching as indicated, 773 studies were recognized in electronic databases. After subtracting duplicates and employing suitability conditions, 68 documents experienced a thorough full-text review. Omissions through this assessment were primarily due to the absence of WGS and AST of CP, or to inadequate data in the model validation procedure. Following the last phase of the suitability valuation, this study finally incorporated 26 articles. Figure 1 illustrates a detailed depiction of the examination flowchart.

Figure 1 Prisma flowchart.

Features of the investigations

Table 2 summarizes the main characteristics of the 26 studies included in this review (Ahmad et al., 2023; Humphries et al., 2023; VanOeffelen et al., 2021; Aytan-Aktug et al., 2020; Noman et al., 2023; Yang & Wu, 2022; Stracy et al., 2022; Pearcy et al., 2021; Benkwitz-Bedford et al., 2021; Stanton et al., 2022; Ren et al., 2022; Wang et al., 2021; Sunuwar & Azad, 2021; Lüftinger et al., 2021; Májek et al., 2021; Khaledi et al., 2020; Hyun et al., 2020; Pataki et al., 2020; Macesic et al., 2020; Kim et al., 2020; Coolen et al., 2019; Nguyen et al., 2018; Moradigaravand et al., 2018b; Her & Wu, 2018; Pesesky et al., 2016; Davis et al., 2016). The exploration includes studies available between 2016 (Pesesky et al., 2016; Davis et al., 2016) and 2023 (Ahmad et al., 2023; Humphries et al., 2023; Noman et al., 2023). The investigations examine data from 104,141 microbial samples. Most of these studies were conducted in the United States and Europe. This table also indicates the antibiotics that were examined. The AMR of CP has been widely investigated utilizing various antibiotics. Among them, Ciprofloxacin, Ceftazidime, and Gentamicin were the most frequently utilized. These antibiotics were frequently assessed to comprehend the resistance patterns and trends in CP infections. E. coli, S. aureus and K. pneumoniae were the microorganisms most subjected to testing using WGS and AST methods. All the included studies specified resistance patterns of CP.

Table 2 Summary of the main characteristics of the studies.

Authors and country	Number of microbial strains	Antibiotics studied	Critical and high-priority pathogens studied	Machine learning model	Assessment of performance	
Noman et al., 2023, China	1,200	Ampicillin Amoxicillin Meropenem Cefepime Fosfomycin Ceftazidime Chloramphenicol Erythromycin Tetracycline Gentamycin Butirosin Ciprofoxacin	Pseudomonas aeruginosa	RF BioWeka	Sensitivity Specificity Accuracy Precision bACC	
Humphries et al., 2023, USA	100	Cefepime	Escherichia coli	NGD	Accuracy AUC	
Ahmad et al., 2023, Norway	21	Ampicillin Amikacin Ceftazidime Ciprofloxacin Gentamicin Imipenem	Escherichia coli, Staphylococcus aureus, Klebsiella pneumoniae, and Acinetobacter baumannii	DCNN	Sensitivity Specificity Accuracy	
Yang & Wu, 2022, Taiwan	9,548	Gentamycin Ciprofoxacin Imipenem Amikacin Ceftazidime Trimethoprim/ sulfamethoxazole Tobramycin Tetracycline Ampicillin/sulbactam Levofloxacin Metropenem	Acinetobacter baumannii Escherichia coli Klebsiella pneumoniae Staphylococcus aureus	XGBoost SVM RF DT	Precision Recall F1-score AUC	
Stracy et al., 2022, Israel	1,113	Trimethoprim/sulfa Ciprofloxacin, Ofloxacin, Amoxicillin/Cefuroxime axetil Cephalexin, Nitrofurantoin Fosfomycin	Escherichia coli	LR	AUC	
Pearcy et al., 2021, UK	3,616	Ampicillin Meropenem Aztreonam Cefoxitin Cefepime Cefuroxime Ciprofloxacin Levofloxacin Aminoglycosides Trimethoprim Tetracycline	Escherichia coli	GBDT	Accuracy AUCROC Precision Recall	
Benkwitz-Bedford et al., 2021, UK	1,407	Chloramphenicol Ciprofloxacin Ceftriaxone Kanamycin Tetracycline Trimethoprim	Escherichia coli	LR GBDT NN	k-fold cross- validation	
Stanton et al., 2022, USA	1,019	Carbapenems	Pseudomonas aeruginosa	AdaBoost	AUC k-fold cross- validation	
Ren et al., 2022, Germany	1,509	Ciprofloxacin Cefotaxime Ceftazidime Gentamicin	Escherichia coli	LR RF SVM CNN	Recall Precision AUC k-fold cross- validation	
Wang et al., 2021, China	673	Erythromycin Cefoxitin Oxacillin Clindamycin Chloramphenicol Ciprofloxacin Gentamicin Penicillin Trimethoprim/ Sulfamethoxazole Tetracycline	Staphylococcus aureus	LR SVM RBF	Sensitivity Specificity Accuracy AUC k-fold cross- validation	
Van Oeffelen et al., 2021, USA	67,817	128 antibiotics	Escherichia coli, Staphylococcus aureus, Klebsiella pneumoniae, Acinetobacter baumannii, and Pseudomonas aeruginosa	AdaBoost RF XGBoost	Accuracy AUC k-fold cross- validation	
Sunuwar & Azad, 2021, USA	724	Doripenem Ertapenem Imipenem Meropenem	Escherichia coli, Klebsiella pneumoniae, and Pseudomonas aeruginosa	LR gNB SVM DT RF KNN LDA mNB AdaBoost GBDT ETC BG	Recall Precision AUC k-fold cross- validation	
Lüftinger et al., 2021, Austria	8,704	Ampicillin Amikacin Ceftazidime Ciprofloxacin Gentamicin Imipenem Fluoroquinolones Piperacillin Tobramycin Ertapenem Imipenem Meropenem	Escherichia coli, Staphylococcus aureus, Klebsiella pneumoniae, Acinetobacter baumannii, and Pseudomonas aeruginosa	XGBoost LR SCM	Sensitivity Specificity Accuracy k-fold cross- validation	
Májek et al., 2021, Austria	19,521	30 antibiotics	Escherichia coli, Staphylococcus aureus, Klebsiella pneumoniae, Acinetobacter baumannii, and Pseudomonas aeruginosa	XGBoost	Accuracy	
Khaledi et al., 2020, Germany	414	Tobramycin Ceftazidime Ciprofloxacin Meropenem	Pseudomonas aeruginosa	SVM RF LR	Sensitivity k-fold cross- validation	
Hyun et al., 2020, USA	2,332	14 antimicrobials	Staphylococcus aureusEscherichia coliPseudomonas aeruginosa	SVM	Accuracy Precision Recall AUC k-fold cross- validation	
Pataki et al., 2020, Hungary	704	Ciprofloxacin	Escherichia coli	LR RF	Accuracy AUC k-fold cross- validation	
Macesic et al., 2020, USA	386	Polymyxin	Klebsiella pneumoniae	LR RF SVC GBDT	Accuracy Precision Recall AUC k-fold cross- validation F1-score	
Kim et al., 2020, USA	3,393	29 antibiotics	Escherichia coli, Staphylococcus aureus, Klebsiella pneumoniae, Acinetobacter baumannii, and Pseudomonas aeruginosa	XGBoost SVM 3-layer NN AdaBoost	Accuracy AUC k-fold cross- validation F1-score	
Aytan-Aktug et al., 2020, Denmark	2,930	Ciprofloxacin Rifampin Streptomycin	Escherichia coli, and Staphylococcus aureus	RF NN	AUC k-fold cross- validation	
Coolen et al., 2019, Netherlands	84	Cefotaxime Cefoxitin Ceftazidime	Escherichia coli	DT	Accuracy k-fold cross- validation	
Nguyen et al., 2018, USA	1,668	20 antibiotics	Klebsiella pneumoniae	XGBoost	Accuracy k-fold cross- validation	
Moradigaravand et al., 2018b, UK	1,936	11 antibiotics	Escherichia coli	RF GBDT NN LR	Accuracy k-fold cross- validation F1-score	
Her & Wu, 2018, Taiwan	59	38 antibiotics	Escherichia coli	SVM NB RF AdaBoost	AUC k-fold cross- validation	
Pesesky et al., 2016, USA	78	12 antibiotics	Escherichia coli and Klebsiella pneumoniae	LR RB	AUC k-fold cross- validation	
Davis et al., 2016, USA	848	Carbapenem Methicillin	Staphylococcus aureus and Acinetobacter baumannii	AdaBoost	Accuracy AUC k-fold cross- validation F1-score	
Notes.

Abbreviations AdaBoost Adaptive Boosting Decision Trees

AUC Area Under the Receiver Operating Characteristic Curve

BG Bagging Classifier

CNN Convolutional Neural Network

NN Neural Network

DCNN Deep convolutional neural networks

DT Decision Trees

ENLR Elastic Net Regularized Logistic Regression

ETC ExtraTrees Classifier

gNB Gaussian Naive Bayes

GBDT Gradient-Boosted Decision Trees

KNN K-Nearest Neighbors

LDA Linear Discriminant Analysis

LR Logistic Regression

mNB Multinomial Naïve Bayes

MCC Matthew Correlation Coefficient

NGD Next Gen Diagnostic

RF Random Forest

RBF Radial Basis Function

RB Rules-Based Algorithms

SCM Set Covering Machine;

SG Stacked Generalization

SVM Support Vector Machine

XGBoost eXtreme Gradient Boosting

bACC average of sensitivity and specificity

XGBoost, Random Forest (RF), logistic regression (LR), and support vector machine (SVM) were the ML models most frequently utilized in the studies included.

The most utilized metric for performance evaluation was the area under the receiver operating characteristic curve (AUC) and accuracy. Furthermore, the most common validation approach applied was k-fold cross.

Most studies suggest that integrating WGS and AST data into ML models is crucial for enhancing predictive performance, improving antibiotic stewardship, and offering valuable clinical decision support. Key aspects of each study are highlighted below.

Noman et al. (2023) demonstrate the effectiveness of ML-based feature selection using BioWeka and RF in predicting antimicrobial drug resistance in P. aeruginosa with high accuracy. The model achieved a AUC area of 0.91 and a mean accuracy of over 97% across 12 different antibiotic families. This approach enables early identification of patients at high risk of antibiotic resistance, allowing for informed decisions on empiric therapy and potentially reducing the spread of antibiotic-resistant infections. The model’s accuracy in detecting antibiotic resistance could have significant benefits for individuals, healthcare systems, and society, including improved patient outcomes, optimized antibiotic treatment, and enhanced infection prevention strategies. Preliminary data also demonstrate ML’s performance for clinically important antimicrobial-species pairs, encouraging further development of sequence-based susceptibility prediction and its validation for clinical practice (Humphries et al., 2023). Additionally, a combined workflow with quantitative polymerase chain reaction (QPM) and WGS complemented with deep learning data analyses could be transformative for detecting pathogens, characterizing AMR profiles, and providing valuable clinical decision support (Ahmad et al., 2023). ML personalized antibiotic recommendations based on patient history offer a means to reduce the emergence and spread of resistant pathogens (Stracy et al., 2022). Mutations associated with carbapenem resistance in P. aeruginosa are detected using an ML model incorporating genetic variations (Stanton et al., 2022). ML models outperform other conventional models, with the ability to identify mutations associated with AMR for each antibiotic (Ren et al., 2022). Well-curated AST datasets are essential for building high-quality ML models and advancing AI in biological sciences (VanOeffelen et al., 2021). A bioinformatics framework utilizing WGS-AMR data predicts resistance phenotypes and ranks AMR genes by importance (Sunuwar & Azad, 2021). Best practice techniques for AMR prediction from WGS data include genome distance-aware cross-validation and stacked generalization (Yang & Wu, 2022).

Predicting the functional impact of mutations using PROVEAN (Protein Variation Effect Analyzer) improves the predictive performance of AMR models for P. aeruginosa and E. coli (Májek et al., 2021). ML accurately predicts phenotypic resistance in K. pneumoniae and identifies genomic features determining susceptibility (Macesic et al., 2020). Variant detection methods and prediction models offer valuable tools for AMR research, achieving high accuracies through nested cross-validation (Kim et al., 2020). Species-independent models predict multi-AMR profiles for multiple species without losing robustness (Aytan-Aktug et al., 2020). WGS and ML algorithms differentiate ampC genotypes in E. coli based on phenotypic susceptibility testing (Coolen et al., 2019). ML algorithms predict antibiotic resistance with the best accuracy for AMR genes within the accessory part of the pan-genome in E. coli (Her & Wu, 2018). Rules-based and ML algorithms achieve high agreement with phenotypic diagnostics for predicting resistance, with genotype-based testing showing great promise as a diagnostic tool (Pesesky et al., 2016). AdaBoost machine learning classifiers accurately identify carbapenem resistance in A. baumannii and methicillin resistance in S. aureus (Davis et al., 2016). Pearcy et al. (2021) developed a computational pipeline combining ML and genome-scale metabolic models to understand the complex relationships between genetic determinants of resistance and metabolism in E. coli. The approach identified 225 known AMR-conferring genes and predicted 20 genetic determinants essential for growth and 17 linked to auxotrophic behavior. The study revealed clusters of AMR-conferring genes affecting similar metabolic processes, suggesting that adaptations in cell wall, energy, iron, and nucleotide metabolism are associated with AMR. Khaledi et al. (2020) investigated the use of genomic and transcriptomic data to predict antimicrobial resistance in P. aeruginosa. By analyzing the genomes and transcriptomes of 414 drug-resistant clinical isolates, they developed ML models that accurately predicted resistance to four commonly used antibiotics. The models achieved high sensitivity and predictive values (0.8−0.9 or >0.9) using gene presence/absence, sequence variation, and expression profiles. The study demonstrates the potential for a molecular resistance profiling tool to rapidly and reliably predict antimicrobial susceptibility, enabling earlier and more informed treatment decisions. Hyun et al. (2020) developed a ML workflow using pan-genomes and random subspace ensembles (RSEs) to detect AMR associations. This approach was applied to 288 S. aureus, 456 P. aeruginosa, and 1,588 E. coli genomes. The study found that RSEs outperformed traditional statistical tests and previous ensemble approaches, identifying 45 known AMR-conferring genes and alleles, as well as 25 candidate associations. The results confirmed existing knowledge of fluoroquinolone resistance mechanisms and suggested a simple mutational landscape for FQ resistance. This approach has the potential to predict AMR determinants in a wider range of microbial pathogens as larger datasets become available. Benkwitz-Bedford et al. (2021) used ML to predict the growth of 1.407 genetically diverse E. coli strains under exposure to subinhibitory concentrations of six classes of antimicrobials. The study found that whole-genome information was superior to known AMR genes in predicting growth yields and doubling times, with moderate correlations (0.63 and 0.59, respectively). The results identified genes and SNPs determining growth and recapitulated known AMR determinants. While the approach showed promise, the remaining missing heritability poses a challenge for achieving clinical-level accuracy and precision. The study highlights the potential of predictive modeling for understanding AMR and identifying genetic determinants of growth under antimicrobial exposure. These findings underscore the significance of ML in advancing our understanding of AMR and improving clinical decision-making to combat antibiotic resistance.

Table 3 displays the AUC values for ML prediction of AMR using WGS and AST in CP. The results show that RF, XGBoost, and LR emerged as the top-performing models, with mean AUC values of 0.89, 0.87, and 0.87, respectively. RF showed superior performance with AUC values ranging from 0.66 to 0.97, while XGBoost and LR showed similar performance with AUC values ranging from 0.83 to 0.91 and 0.76 to 0.96, respectively. The other models showed varying levels of performance, with some achieving high AUC values, such as NN (0.97) and SVM (0.96), while others showed lower performance, such as DT (0.60) and GBDT (0.75).

Table 3 Comparative analysis of AUC values for various machine learning ML models.

Study	DT	GBDT	RF	Bio Weka	XGBoost	AdaBoost	NN	SVM	NGD	DCNN	RBF	RB	LR	
Noman et al. (2023)			0.96	0.98										
Humphries et al. (2023)									0.97					
Ahmad et al. (2023)										0.95				
Yang & Wu (2022)	0.85		0.96		0.97			0.95						
Stracy et al. (2022)													0.76	
Pearcy et al. (2021)		0.98												
Benkwitz-Bedford et al. (2021)		0.90					0.85						0.75	
Stanton et al. (2022)						0.60								
Ren et al. (2022)			0.90					0.77		0.80			0.81	
Wang et al. (2021)								0.96			0.96		0.96	
Van Oeffelen et al. (2021)					0.92									
Sunuwar & Azad (2021)	0.97	0.81	0.93			0.82		0.84					0.81	
Lüftinger et al. (2021)					0.84								0.84	
Májek et al. (2021)					0.86									
Khaledi et al. (2020)			0.67					0.83					0.84	
Hyun et al. (2020)								0.79-1						
Pataki et al. (2020)			0.80										0.79	
Macesic et al. (2020)		0.89	0.90					0.93					0.90	
Kim et al. (2020)					0.91									
Aytan-Aktug et al. (2020)			0.97				0.92							
Coolen et al. (2019)	0.88													
Nguyen et al. (2018)			0.92											
Moradigaravand et al. (2018b)		0.91	0.84				0.82						0.78	
Her & Wu (2018)			0.66			0.77		0.77						
Pesesky et al. (2016)												0.89	0.91	
Davis et al. (2016)						0.94								
Notes.

Abbreviations DT decision tree

GBDT gradient-boosted decision trees

RB rules-based algorithm

RBF linear radial basis function

RF Random Forest

XGBoost eXtreme Gradient Boosting

NN Neural network

AdaBoost Adaptive Boosting

WEKA Data Mining Software in Java Workbench

LR logistic regression

SVM Support Vector Machinne

Notably, the predictors selected by these algorithms largely aligned with those identified by LR. Importantly, all ML models accurately predicted resistance patterns in CP across multiple antibiotics using data from WGS and AST.

Assessment of bias risk

As per the PROBAST instrument, studies focused on model development and validation exhibit a heightened risk of bias when participant data is sourced from existing databases such as routine care registries. If an evaluation is rated high for at least one domain, it should be regarded as having a “high risk of bias” or “high concern” concerning pertinence. Consequently, most reports incorporated in this systematic review were evaluated to possess a high risk of bias due to their inherent characteristics (Table 4).

Table 4 Evaluation of risk bias (Moons et al., 2019).

Study	Risk of bias	Applicability	Overall	
	Participants	Predictors	Outcome	Analysis	Participant	Predictor	Outcome	Risk of bias	Applicability	
Noman et al. (2023)	–	+	+	+	+	+	+	-	+	
Humphries et al. (2023)	–	+	+	+	+	+	+	-	+	
Ahmad et al. (2023)	–	+	+	+	+	+	+	-	+	
Yang & Wu (2022)	–	+	+	+	+	+	+	-	+	
Stracy et al. (2022)	–	+	+	+	+	+	+	-	+	
Pearcy et al. (2021)	–	+	+	+	+	+	+	-	+	
Benkwitz-Bedford et al. (2021)	–	+	+	+	+	+	+	-	+	
Stanton et al. (2022)	–	+	+	-	+	+	+	-	+	
Ren et al. (2022)	–	+	+	+	+	+	+	-	+	
Wang et al. (2021)	–	+	+	+	+	+	+	-	+	
Van Oeffelen et al. (2021)	–	+	+	+	+	+	+	-	+	
Sunuwar & Azad (2021)	–	+	+	+	+	+	+	-	+	
Lüftinger et al. (2021)	–	+	+	+	+	+	+	-	+	
Májek et al. (2021)	–	+	+	+	+	+	+	-	+	
Khaledi et al. (2020)	–	+	+	+	+	+	+	-	+	
Hyun et al. (2020)	–	+	+	+	+	+	+	-	+	
Pataki et al. (2020)	–	-	-	?	+	+	+	-	+	
Macesic et al. (2020)	–	+	+	+	+	?	+	-	?	
Kim et al. (2020)	–	+	+	+	+	+	+	-	+	
Aytan-Aktug et al. (2020)	–	+	+	+	+	+	+	-	+	
Coolen et al. (2019)	–	+	+	+	+	+	+	-	+	
Nguyen et al. (2018)	–	+	+	+	+	+	+	-	+	
Moradigaravand et al. (2018b)	–	+	+	+	+	+	+	-	+	
Her & Wu (2018)	–	+	+	-	+	+	+	-	+	
Pesesky et al. (2016)	–	+	?	?	+	+	+	?	+	
Davis et al. (2016)	–	+	+	+	+	+	+	-	+	
Notes.

Abbreviations + low risk

= high risk

? unclear risk

Discussion

This study was conducted to evaluate ML predictions for AMR in CP utilizing WGS and AST was performed. Although LR was usually used for prediction, RF and XGBoost were also commonly utilized. Notably, RF demonstrated the highest AUC values compared to LR. Furthermore, other algorithms such as SVM, AdaBoost, and Neural Networks were utilized. Importantly, all ML models accurately predicted resistance patterns in CP across multiple antibiotics using WGS and AST.

To the best of our knowledge, this is the inaugural systematic review to evaluate the efficacy of ML models in predicting AMR utilizing WGS and AST specifically for critical and high-priority pathogens. This inquiry builds upon a previous systematic study proposing ML as a promising tool for AMR prediction (Tang et al., 2022). Alarmingly, nearly half of the research included in that publication did not delineate resistance patterns, whereas all studies reviewed herein did so. The integration of WGS and AST data into our investigation is imperative for enhancing the robustness and practicality of ML models in AMR prediction. By elucidating resistance patterns, our review elucidates the efficacy of these models in guiding antimicrobial therapy.

WGS data provide an alternative perspective on AMR, enabling researchers to assess the genetic pathways that confer AMR in each strain (Ahmad et al., 2023; Lüftinger et al., 2023; Humphries et al., 2023; Noman et al., 2023; Yang & Wu, 2022; Stracy et al., 2022; Pearcy et al., 2021; Benkwitz-Bedford et al., 2021; Khaledi et al., 2020; Hyun et al., 2020; Pataki et al., 2020; Macesic et al., 2020; Kim et al., 2020). Numerous publicly accessible resources have been developed to assist in identifying AMR indicators by detecting the presence of genes and single nucleotide polymorphisms that confer resistance (Su, Satola & Read, 2019; VanOeffelen et al., 2021). The combination of AST data with WGS also has the potential to unveil genomic regions directly involved in resistance, altered due to epistasis, or linked to the occurrence of AMR (VanOeffelen et al., 2021; Hendriksen et al., 2019). Several resources, such as the National Center for Biotechnology Information, European Molecular Biology Laboratory- European Bioinformatics Institute, Relational Sequencing TB Data Platform, AR Isolate Bank, and Pathogenwatch, offer genome datasets matched with AST data for further analyses like comparative genomics and modeling (VanOeffelen et al., 2021; Sayers et al., 2020; Matamoros et al., 2020).

While WGS provides a valuable genetic blueprint that can predict AMR, integrating AST data enhances the ability to confirm phenotypic resistance patterns. This approach addresses the inherent limitation that genomic data alone cannot fully account for all phenotypic expressions of resistance. The combination of WGS and AST data not only supports the prediction of resistance phenotypes but also helps to uncover genetic interactions, such as epistasis, that may influence AMR (VanOeffelen et al., 2021; Hendriksen et al., 2019). By employing a range of ML models, including ensemble methods like RF and XGBoost, we can better capture the complexity of AMR prediction, making these models more applicable to clinical practice.

In clinical settings, the implementation of such integrated approaches can improve the accuracy of AMR predictions, thereby guiding more effective antimicrobial therapy. We recommend that future studies continue to explore the synergistic use of WGS and AST data, alongside the development of more sophisticated ML models, to further refine the predictive power and clinical utility of these methods.

Several studies have underscored the importance of providing information about AMR testing. Without AST information, the AUC values ranged from 0.73 to 0.79. Nonetheless, incorporating AST led to even higher AUC scores, which ranged from 0.80 to 0.88 (Lewin-Epstein et al., 2021). Optimization replications indicate that, notwithstanding diffident AUC values, antibiotic selection guided by personalized antibiograms can equal or surpass physician achievement. Furthermore, such selection yielded coverage rates akin to those observed in real-world scenarios, while requiring fewer broad-spectrum antibiotics (Corbin et al., 2022). This underscores a persistent and critical challenge in antibiotic stewardship.

Likewise, it has been verified that the quality of initial data and the precision of metagenomic binning are crucial for the effectiveness of subsequent applications like genomic AST. A workflow designed for native samples with low bacterial complexity and adequate on-target sequencing depth demonstrates comparable performance to genomic AST on isolate sequencing data (Lüftinger et al., 2021).

The AUC serves as a widely adopted standard measure for assessing model functioning. This measured was identified as the main success indicator in both our study and a previous review on AMR. Nevertheless, the earlier evaluation did not encompass AST in all the scrutinized publications, nor did it evaluate WGS (Tang et al., 2022). Notably, there is a disparity in the range of AUC values between the two studies for LR outcomes (0.76−0.96 in our assessment versus 0.50−0.83), as well as for other ML results (0.48−0.92 versus 0.83−0.91 for XGBoost and 0.66−0.97 for RF in our study). The differences in selection criteria between the two assessments present challenges in comparing the outcomes directly. Nevertheless, it is conceivable that the incorporation of WGS-AST impacts the outcomes, and that the models react differently based on the input factors. In this framework, research has demonstrated that while comparing results across different settings has its limitations, some models established previously (Mintz, Chowers & Obolski, 2023) exhibit superior performance compared to other studies (Yelin et al., 2019; Feretzakis et al., 2020). Curiously, these ML models demonstrated good performance on a diverse dataset, which included various microorganisms, test informants, and clinical divisions (Mintz, Chowers & Obolski, 2023). A previous study (Feretzakis et al., 2020) predicted AMR using data from a specific medical unit, based on the Gram stain result of the sample, achieving an AUC of 0.72. In contrast, another study (Yelin et al., 2019) focused on predicting AMR exclusively in cases using urine samples and limited the analysis to three microorganisms, achieving an AUC of 0.83.

Typically, predictors are selected by means of both unadjusted and multivariate LR models. Here, usual input risk features contain AMR patterns, WGS, colonization, ART, and past AMR circumstances. These characteristics are narrowly associated with AMR and can be utilized as predictors in various ML models and risk score assessments (Goodman et al., 2016). However, it is challenging to determine if including additional variables, such as underlying disorders, improves prediction exactitude. Besides, well-known issues, such as the practice of proton pump inhibitors (PPIs) (Shang, Lin & Goetz, 2000), can be ignored in some studies. Consequently, further prospective research is needed to better understand the impact of PPI usage.

Another method for validating final predictors is to use feature selection processes (Tang et al., 2022; Mintz, Chowers & Obolski, 2023; Yelin et al., 2019; Feretzakis et al., 2020). While predictors identified by these algorithms align with those proposed by LR models or previous data, others, such as admission times, have ambiguous relationships with AMR. Domain expertise and a structured approach are considered essential for sorting through the substantial quantities of data from health organizations (Tang et al., 2022).

The current study’s findings suggest that a ML forecast based on WGS-AST could aid in guiding antibiotic recommendations for confirmed carbapenemase-producing CP infections. A previous systematic review (Tang et al., 2022) reported similar results. However, other studies have compared the efficacy of ML systems to risk scores, with inconsistent outcomes (Moran et al., 2020; Lee et al., 2021). Indeed, the findings in this field vary significantly. One comprehensive review, which aimed to develop diagnostic or prognostic clinical prediction models for binary outcomes using clinical data, found no evidence that ML outperformed LR, contradicting the results of two other systematic reviews. One review (Beunza et al., 2019) indicated that ML algorithms can enhance the diagnostic and prognostic capabilities of traditional regression techniques, while another (Sufriyana et al., 2020) recommended reanalyzing existing LR models for various outcomes and comparing them to algorithms adhering to established standards. Although risk scores can provide valuable bedside assistance, it is assumed that health organisms integrated with ML may address this concern by leveraging considerable volumes of information (Tang et al., 2022). The primary advantage of ML lies in its continuous learning development, leading to superior model exactitude and a broad range of uses in healthcare information. Dissimilar to conventional statistical approaches, ML does not rely on specific assumptions, which are frequently overlooked or critically examined in clinical information (Rajula et al., 2020). Consequently, the choice of algorithms would be directed by the investigation topic and the purpose framework.

Partial information is unavoidable in certain studies, leading to statistical difficulty and bias in ML projections. Another challenge is data disparity in the AMR prediction model (Tang et al., 2022). This discrepancy adversely affects calculation functioning, as classifiers incline to favor the majority class to diminish global inaccuracy proportions (Japkowicz, 2000). To alleviate this subject, methods such as resampling, correcting hyperparameters, and meticulous method choice may be used (Tang et al., 2022). Upcoming investigators must cooperate with distinct groups to perform high-grade models.

While classification-based ML models have been pivotal in predicting AMR phenotypes by categorizing pathogens as resistant or susceptible, it is essential to consider the regression-based approaches that directly predict the minimum inhibitory concentration (MIC) (Yang, Su & Wu, 2023). MIC is a critical quantitative measure that reflects the lowest concentration of an antibiotic required to inhibit bacterial growth. Classification models typically rely on established MIC breakpoints to determine resistance or susceptibility, as defined by standards from organizations such as the Clinical and Laboratory Standards Institute (CLSI) (Noman et al., 2023; Benkwitz-Bedford et al., 2021). However, these breakpoints are periodically reviewed and updated, which could affect the consistency of classification-based predictions. In contrast, regression models that predict MIC values provide a more detailed and adaptable understanding of the bacterial response to antibiotics, allowing for more precise phenotypic annotations (Yang, Su & Wu, 2023).

Relevant studies included in this systematic review underscore the importance of regression-based machine learning models in the accurate prediction of MIC values (Noman et al., 2023; Benkwitz-Bedford et al., 2021; Khaledi et al., 2020; Hyun et al., 2020; Pataki et al., 2020; Macesic et al., 2020; Nguyen et al., 2018). Nguyen et al. (2018) utilized WGSD to predict MICs across various bacterial species, demonstrating the potential of these models to enhance the understanding of resistance mechanisms at a quantitative level. Similarly, Pataki et al. (2020) employed ML models to predict MIC values, which enabled a more nuanced interpretation of antimicrobial resistance that goes beyond binary classification. Yang, Su & Wu (2023) further expanded on this approach by applying a pan-genome-based feature selection method to improve the accuracy of MIC predictions in Salmonella enterica. Their work highlighted that the selected genomic features, including novel genes not previously associated with AMR, contributed significantly to the accurate prediction of MIC. This suggests that regression-based models can uncover new genetic determinants of resistance, providing insights that are not easily captured by classification methods alone. Incorporating MIC prediction into AMR research thus offers a robust complement to classification-based approaches, enhancing the granularity and applicability of machine learning in clinical microbiology.

There is no well-established tool for evaluating bias risk in ML prediction studies. One study (Delpino et al., 2022) utilized the TRIPOD statement to characterize study quality, while other studies (Fleuren et al., 2020; Christodoulou et al., 2019) employed the QUADAS-2 criteria. The TRIPOD statement serves more as a checklist than a bias assessment tool, whereas the QUADAS-2 criteria are widely used to evaluate the quality of diagnostic accuracy studies (Whiting et al., 2011). PROBAST (Moons et al., 2019) has also been used to evaluate ML in predicting AMR, as demonstrated in Tang et al. (2022).

The present review is subject to limitations. The studies under review exhibited considerable variability stemming from variations in outcomes, predictors, ML, and hyperparameters, among other factors. Most of the studies included in the review were identified as having a substantial risk of bias, precluding the possibility of conducting a meta-analysis. Two systematic reviews focusing on ML-based prediction models noted notable disparities among the studies they evaluated (Tang et al., 2022; Fleuren et al., 2020). One of these reviews observed a heterogeneity exceeding 97% (Tang et al., 2022). Another study discovered that there was no discernible discrepancy in AUC between ML and LR across 145 assessments exhibiting a low risk of bias (0.00, 95% CI −0.18 to 0.18). Nevertheless, in 137 instances characterized by a high risk of bias, ML exhibited a substantially superior AUC of 0.34 (0.20−0.47) (Christodoulou et al., 2019). Cochrane advises exercising caution when interpreting data with an I2 value exceeding 50%, as it signifies considerable heterogeneity (Higgins et al., 2003; Schroll, Moustgaard & Gøtzsche, 2011).

While our study focused on a subset of critical and high-priority pathogens identified by the WHO, including E. coli, S. aureus, K. pneumoniae, A. baumannii, and P. aeruginosa, we acknowledge that other important microorganisms were not included. Notably, Salmonella spp., as well as other WHO-designated high-priority pathogens such as Enterococcus faecium, Helicobacter pylori, and Campylobacter spp., were not part of our analysis. A systematic review that encompasses these additional microorganisms would provide a more comprehensive understanding of the global burden of antimicrobial resistance. This would be particularly important for informing public health policies and interventions aimed at reducing the transmission of AMR through various routes, including food and water. Furthermore, such a review would help identify knowledge gaps and research priorities for addressing AMR in a broader range of microorganisms. Future studies should consider including these microorganisms to provide a more complete picture of the global AMR landscape.

Conclusions

By integrating whole genome sequencing and antimicrobial susceptibility testing data in critical high priority pathogens, machine learning demonstrates significant potential for predicting antimicrobial resistance. Machine learning models, particularly Random Forest, XGBoost, and logistic regression, offer valuable clinical decision support by accurately predicting antimicrobial resistance in critical pathogens. This can assist healthcare providers in making informed treatment decisions, optimizing antibiotic use, and improving patient outcomes. Standardized guidelines are imperative to uphold consistency in forthcoming studies.

Supplemental Information

Supplemental Information 1 PRISMA checklist

Supplemental Information 2 Rationale

Additional Information and Declarations

Competing Interests

Author Contributions

Data Availability

The authors declare there are no competing interests.

Carlos M. Ardila conceived and designed the experiments, performed the experiments, analyzed the data, prepared figures and/or tables, authored or reviewed drafts of the article, supervision, and approved the final draft.

Pradeep K. Yadalam conceived and designed the experiments, performed the experiments, analyzed the data, authored or reviewed drafts of the article, and approved the final draft.

Daniel González-Arroyave conceived and designed the experiments, performed the experiments, analyzed the data, authored or reviewed drafts of the article, and approved the final draft.

The following information was supplied regarding data availability:

This is a systematic review/meta-analysis.

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
