# Peer review of "Integrating whole genome sequencing and machine learning for predicting antimicrobial resistance in critical pathogens: a systematic review of antimicrobial susceptibility tests"

_PeerJ, doi:10.7717/peerj.18213_

## Round 0.1 · original submission · Major Revisions

Although one of the reviewers recommended rejection, I decided to give you an opportunity to reply to the critiques and to revise the manuscript. Please address all the issues pointed by both reviewers and mend manuscript accordingly.

Reviewer 1 ·

Basic reporting

This manuscript, titled “Integrating Whole Genome Sequencing and Machine Learning for Predicting Antimicrobial Resistance in Critical Pathogens: A Systematic Review of Antimicrobial Susceptibility Tests,” presents tabulated and analyzed data on whole genomes and metagenomes from several journals that utilize machine learning. This systematic review reveals a new method for predicting antimicrobial-resistant pathogens. The recommended models mentioned in this paper can serve as guidelines for future studies. However, a few confusing aspects were identified. I recommend the publication of this manuscript in PeerJ after revision according to the comments listed.

Experimental design

Study Selection (lines 155-163): The authors mentioned that 1-3 investigators selected the journals for analysis, even though kappa statistics were applied. Relying on such a small number of investigators can introduce subjectivity. The authors should provide a more detailed explanation of this limitation.

Validity of the findings

According to this systematic review, whole genome sequencing (WGS) and machine learning can help predict antimicrobial resistance in critical pathogenic bacteria without the need for prolonged culturing. However, it is important to note that genomic data cannot fully confirm phenotypic characteristics. The discussion section should address this gap and provide recommendations on how genomic methods can be effectively applied in clinical practice such as by using more than one predictive model.

Additional comments

Several critical pathogens are mentioned in this study, including Escherichia coli, Staphylococcus aureus, Klebsiella pneumoniae, Acinetobacter baumannii, and Pseudomonas aeruginosa. However, I noticed that Salmonella spp. were not included. Salmonella is globally recognized as one of the most serious foodborne pathogens and a significant cause of antimicrobial resistance (AMR). The authors should provide a brief explanation for the exclusion of Salmonella in this study.

Annotated reviews are not available for download in order to protect the identity of reviewers who chose to remain anonymous.

Reviewer 2 ·

Basic reporting

The manuscript is about a systematic review of whole-genome-based machine learning prediction and development of antimicrobial susceptibility test. I however find severe flaws in this review, including the unreasonable mix-up of genomics and metagenomics and secondly the authors seem to not read the papers at all. I also find severe lack of literatures in this review. Below please find my opinions.

1. I wonder whether the authors have really read the paper. For example, citation [20] is about a pipeline recovering the pathogens and their antimicrobial resistance patterns; it is not really a machine learning paper. Yes at the end of the paper they utilized machine learning (XGBoost) to compare their approach with individual genome-based methods; however the latter is really not the point of the paper. Another example is citation [2], which is about the "simulation" of 2000 artificial metagenomes from known bacteria to test the efficacy of their proposed optimized assembly and binning process. The machine learning part is again not the point, not to mention that the "2000 microbiome samples" listed in table 2 are not real microbiome samples. I urge the authors to focus on just the machine learning papers by at least read the papers.

2. There should be way more papers than just 20. For example, a very well-written paper published by Khaledi et al. on EMBO Mol Med 2020 (doi:10.15252/emmm.201910264) was not included and discussed at all. Another example is Hyun et al. PLOS Computational Biology 2020 (doi:10.1371/journal.pcbi.1007608). These two papers are worth investigating for the authors' purpose since they are mainly focused on machine learning AMR prediction. Please don't just add these two papers into the collection but instead optimize the search process to mine more relevant papers and really read them to ensure they are applicable.

Experimental design

1. The authors should re-design their search criteria to include more adequate papers and really read the papers.

Validity of the findings

1. It does not make sense to mix microbiome/metagenomics problems with culture-based genomic data, as the nature of these two types of data are very different from each other. For example, while culture-based genomics data can be used to train and evaluate the machine learning models, metagenomics data cannot fulfill this goal since there are simply too many species without uncharacterized traits in the metagenomes. In other words, only culture-based genomics data can be used to evaluate the machine learning methods instead of metagenomics samples.

2. It also does not make sense to hear that the authors describe "105,414 microbiome samples" were identified. Firstly microbiome samples cannot be used to evaluate gene/genome-based machine learning methods (see above reason), and secondly it is a bit surprising to see that there are so many "microbiome" samples. I wonder whether the author mix both individual genome and metagenome into their so-called "microbiome". For example, in the first paper of table 2, there are actually 1200 individual genomes instead of microbiome samples used to validate the developed method. So very likely the authors did not know the differences between culture-based genomics and metagenomics and just mix them up.

Additional comments

1. Talking about susceptibility testing, while the authors focus on classification-based machine learning processes, there should be one more thing to be discussed in the review: the prediction of minimum inhibitory concentration (MIC). There are several papers that utilize regression-based machine learning models to predict MIC values, including Nguyen et al. Sci Rep 2018 (doi:10.1038/s41598-017-18972-w), Pataki et al. Sci Rep. 2020 (citation [28] of the manuscript, in which the authors included but not discussed at all), and Yang et al. Frontiers in Genetics 2023 (doi:10.3389/fgene.2023.1054032). This is because the AMR phenotype is usually translated from MIC via public standards published by institutes such as CLSI. The discussio and comparison of the classification- and regression-based methods should be one of the points of the review.

---

## Round 0.2 · accepted · Accept

All concerns of the reviewers were adequately addressed and revised manuscript is acceptable now. Two remaining recommendations of the reviewer #2 should be fixed at the proof stage.

Reviewer 2 ·

Basic reporting

1. The single paragraph spanning four pages from line 226-300 is too long. Please consider splitting the paragraph into several paragraphs, in which each paragraph cover a sub-topic of the literature description.

2. Should be AUROC (or AUC) instead of AuROC. Note the uppercase U instead of lowercase.

Experimental design

The authors addressed by concerns. I have no other comments.

Validity of the findings

The authors addressed by concerns. I have no other comments.

Additional comments

The authors addressed by concerns. I have no other comments.